# Systematic review on e-cigarette and its effects on weight gain and adipocytes

**Rafidah Hod, Nurul Huda Mohd Nor, Sandra Maniam** *

Department of Human Anatomy, Faculty of Medicine and Health Sciences, Universiti Putra Malaysia, Selangor, Malaysia

* sandra@upm.edu.my

**Data Availability Statement:** All relevant data are within the paper and its Supporting Information files.

**Funding:** The author(s) received no specific funding for this work.

## Abstract

Smoking and obesity are leading causes of morbidity and mortality worldwide. E-cigarette which was first introduced in 2000s is perceived as an effective alternative to conventional tobacco smoking. Limited knowledge is available regarding the risks and benefits of e-cigarettes. This study systematically reviews the current literature on the effects of e-cigarettes on body weight changes and adipocytes. The search was performed using OVID Medline and Scopus databases and studies meeting the inclusion criteria were independently assessed. This review included all English language, empirical quantitative and qualitative papers that investigated the effects of e-cigarettes on bodyweight or lipid accumulation or adipocytes. Literature searches identified 4965 references. After removing duplicates and screening for eligibility, thirteen references which involve human, in vivo and in vitro studies were reviewed and appraised. High prevalence of e-cigarette was reported in majority of the cross sectional studies conducted among respondent who are obese or overweight. More conclusive findings were identified in in vivo studies with e-cigarette causing weight decrease. However, these observations were not supported by in vitro data. Hence, the effect of e-cigarette on body weight changes warrants further investigations. Well-designed population and molecular studies are needed to further elucidate the role of e-cigarettes in obesity.

## Introduction

Prevalence of nicotine use among obese population is high and it was reported that life expectancy of an obese smoker is 13 years less than that of a normal-weight non-smoker [1]. Smoking cessation results in considerable improvements in health, however it is often accompanied by weight gain. Obesity and cigarette smoking are the leading preventable causes of death in developed societies. Both obesity and cigarette smoking are also important risk factors in many age-related diseases, accelerating the aging process via increasing oxidative stress and inflammation. Increase in body weight was observed upon smoking cessation among adults with stronger nicotine dependency resulting in higher weight gain [2–6].

The majority of tobacco use was reported to start during youth or adolescence [7]. An increase by 5% was noted in the prevalence of vaping or e-cigarette in the past 30 days among

**Competing interests:** The authors have declared that no competing interests exist.

adolescents aged 16–19 years in the US and Canada in 2018 [8]. The increase was also noted in the daily use of e-cigarette among US young adults in 2014–2018. However the prevalence among older adults remained stable or declined [9]. Similarly, in Myanmar, the use of e-cigarette was reported to be significantly higher among male young adults aged 18–29 years old who are heavy smokers [10].

The epidemiology data which shows the association between smoking and weight gain has been known for many years. Smoking cessation is associated with a mean increase of 4–5 kg in body weight after 12 months of abstinence, and most weight gain occurs within three months of quitting [11, 12]. Nicotine cessation in rats resulted to hyperphagia that caused increased body weight which was achieved substantially via an increase in meal size consumed by the rats [13]. Conversely, nicotine exposure reduced high fat diet-induced weight gain [14–16] that further led to decrease in both fat and lean body mass in mice [17]. Nicotine was found to significantly increase the production of ROS [18–22] and induce phosphorylation of AMPK. Activation of AMPK is associated with metabolic effect of nicotine which includes inhibition of lipogenesis and fat oxidation that is linked to weight loss [16, 23, 24].

E-cigarettes are battery-powered devices that produce aerosols by heating a liquid solution (with or without nicotine) with a metal coil. "E-cigs", "e-hookahs", "vapes", "vape pens", "ciga-likes", "mods", "JUUL", "tank systems", "electronic vapour product" and "electronic nicotine delivery systems (ENDS)" are all terms used to describe e-cigarettes. Some e-cigarettes have the appearance of traditional cigarette sticks, cigars, or pipes. Some of them resemble USB flash drives, pencils and other commonplace goods [25]. Prevalence of current e-cigarette users ranges from 0.9% [26], 3.1% [27], 3.2% [28], and 7.6% [9], in China, Poland, Malaysia and South Korea respectively. Interestingly, the prevalence e-cigarette use among youth was associated with smoking status [29].

E-cigarettes were first introduced to the US market in the mid-2000s [30] and its expenditure and online engagement through social media have increased over time, frequently framed as a tobacco-free alternative [31]. Despite cautionary statements by FDA regarding e-cigarettes, studies often found that perceptions on the benefits of e-cigarettes among various populations remain. In adults, some of the perceived benefits of e-cigarettes among adult e-cigarette users are i) less addictive and less likely to cause cancer than traditional cigarettes, ii) increases concentration, iii) weight control, iv) safety of ingredient liquids, v) safer for the environment and bystanders and vi)smoking cessation purposes [32]. Smoking cessation is the most common reason for e-cigarette use. However, other reasons include pleasant taste and multi flavours, reduced stress and weight gain control, avoidance of smoking restrictions by the dual use of tobacco products and e-cigarettes, product convenience, curiosity, and social environment influences [32]. In vitro studies have shown that most of the harm to bronchial epithelial cells arose from the volatile component of cigarette smoke compared to total particulate matter or nicotine [33, 34].

There is moderate evidence that e-cigarettes with nicotine increase nicotine cessation rates compared to nicotine replacement therapy [35]. E-cigarettes is one of the most popular aid to nicotine smoking cessation in the UK and has been proposed that it may reduce weight gain [36, 37]. To date, there is no strong evidence supporting the hypothesis that e-cigarettes reduce weight gain. This study aims to collate and integrate the literature assessing the use of e-cigarette associated with body weight changes in various study subjects.

## Materials and methods

The details regarding search inclusion and exclusion are illustrated in Fig 1. The search terms 'e-cigarettes' OR 'electronic cigarettes' OR 'vapor cigarettes' OR 'vapes' OR 'electronic nicotine

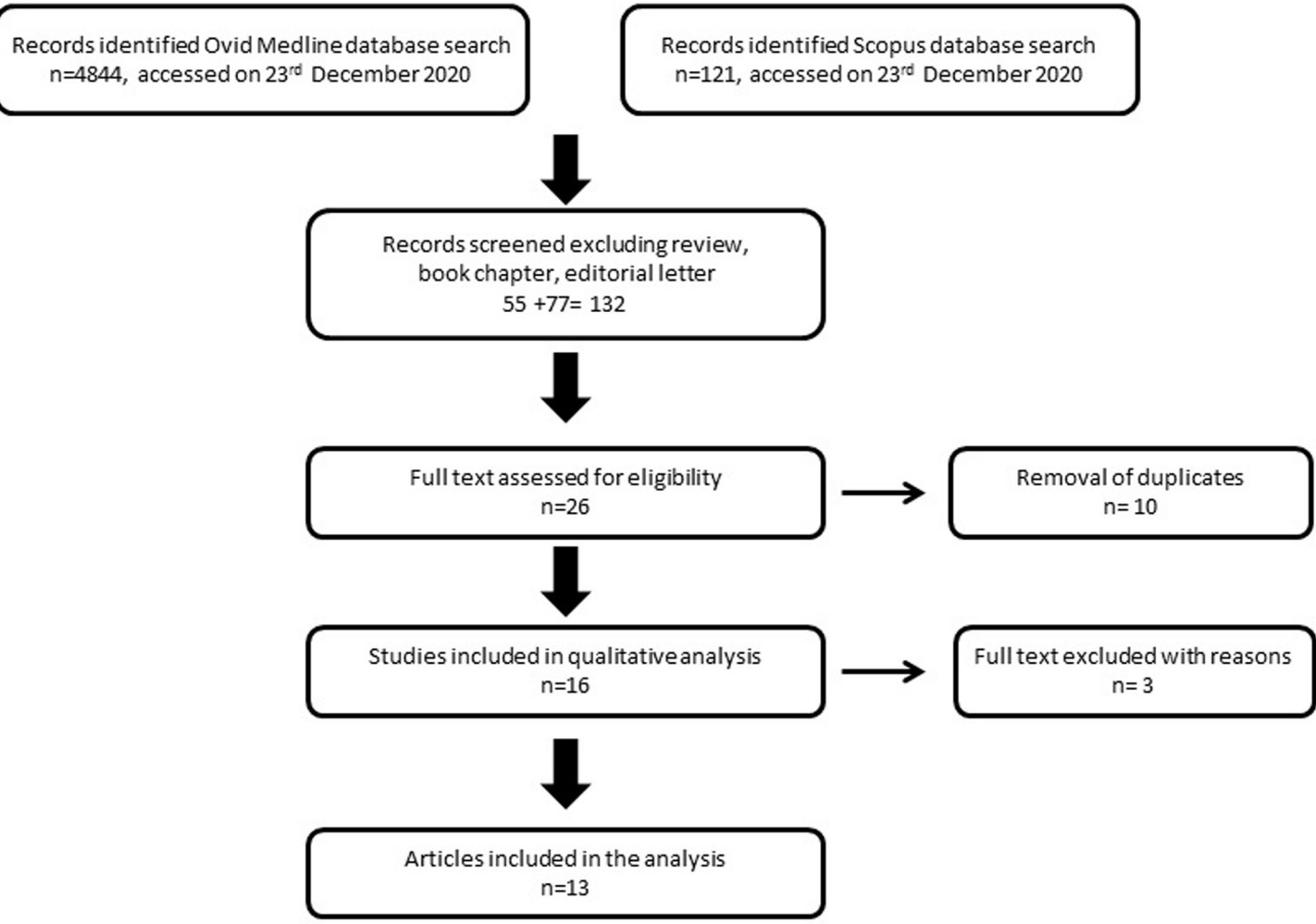

**Fig 1. Flowchart of the selection process.**

delivery device' OR 'juul' AND 'obesity' OR 'overweight' OR 'fat' OR 'obese' OR 'unhealthy weight' OR 'high BMI' were used to explore the primary topic which was to describe the effects of e-cigarette on weight gain using the OVID MEDLINE databases (121) and Scopus database (4,844) accessed on 23rd December 2020. Abstracts were screened and reviewed and abstracts with the following criteria were excluded: (i) non-English abstracts; (ii) do not have any direct relevance to the search topic of weight gain and e-cigarette (iii) review articles. After manual screening, 132 articles were identified. Following the removal of duplicates, 26 articles were shortlisted for full-text article assessment. Sixteen articles met the inclusion criteria and 3 were excluded. The excluded articles involve one was review article and two articles that did not report any information on the effect of–cigarette to weight gain. The search and identification of articles that met the inclusion criteria were performed simultaneously by all authors. The data extraction and quality of bias assessment were performed independently by each author using a standardized data extraction form agreed by all reviewers. Discrepancies following independent data abstraction and quality assessment were resolved until 100% agreement was achieved. Studies include publications ranging from the year 2015–2021. This protocol is registered with PROSPERO with ID CRD42021238704.

## Study quality/risk of bias ratings

Risk of bias ratings were assigned to each ratings of the epidemiology, experimental animal, and in vitro studies for each of their respective domains; study relevance is assessed during the evidence integration process. Risk of bias was evaluated at individual level which means that one individual study will receive multiple risk of bias evaluations by all reviewers. Summaries of risk of bias were presented through heatmaps for each study (S1 Table).

The guidance from the OHAT Handbook [38] for dividing studies into three tiers of study quality. In this approach, studies are divided into tiers of decreasing quality as follows:

Tier 1 –A study must be rated as "definitely low" or "probably low" risk of bias for all key domains for that study type AND have most (i.e. at least half) other risk of bias domains as "definitely low" or "probably low."

Tier 2 –Study does not meet the criteria for Tier 1 or Tier 3.

Tier 3 –A study must be rated as "definitely high" or "probably high" risk of bias for all key domains for that study type AND have most (i.e. at least half) other risk of bias domains as "definitely high" or "probably high."

As stated in the in the OHAT Handbook [38], Tier 3 studies should not be included in the integration of evidence. All studies included in this study were Tier 1 and 2.

# Results

## Study characteristics

This review evaluates various studies associated with obesity and the effects of e-cigarette which were noted in 10 human, two in vivo and one in vitro study. Year of publication ranged from 2015 to 2020.

All human studies were cross sectional except one which was longitudinal [39]. The human studies were carried in a range of countries; Korea [40, 41], Hawaii [42], China [43] and US [39, 44–48] and the data were either extracted from annual national survey [40, 41], survey conducted at district level [39, 43, 44], among undergraduate students in university [42, 47] and from survey conducted in other places such as workplace [48] and hospital [46]. Majority of study included all ethnicity except for one which exclusively focused on American Indian population [46]. The total number of participants varied from 452 to 17,656 with majority of studies included both gender.

The two in vivo studies included in this review explored the effects of e-cigarette in mouse models. One of the studies measured the effect of e-cigarette on adult male mice [49] whilst the other study used pregnant mice to investigate the in utero effect of e-cigarette in the adult male mice, dams and pups [50]. Adipocytes derived from mouse embryo (3T3-L1 preadipocytes) were used in the in vitro study to observe nicotine-mediated adipogenic differentiation [51].

## Study subjects

Demographic differences such as age is significantly associated with substance abuse behaviour [52]. The ten human studies evaluated in this review recruited participants from various age groups. The youngest age range that was more than 13 years old [43]. Several studies focused on younger adults aged more than 20 years old [40] and 19 years old [41] and four studies recruited respondents aged more than 18 years old [39, 46–48]. Three other studies used a more defined age range as an inclusion criterion which includes 12–19 years old [44], 15–18

years old [45] and 18–25 years old [42]. The demographic data of human studies highlighted in this review is summarized in Table 1. Adolescence was shown to have high prevalence in obesity [53] and early experimentation of addictive substance use which includes tobacco [54]. Studies [39, 40, 46, 47] that used a wider age group as an inclusion criterion includes respondents from early adolescent to elderly (above 65 years old [55]).

## Measuring e-cigarette consumption

In this review, five studies measured e-cigarette prevalence for the past 30 days as current users [41–45] whilst two studies emphasized on e-cigarette experimentation by measuring if the respondent have ever tried e-cigarette [39, 46] and another study which seeks the respondents' perception as a regular user of e-cigarette [47]. More detailed information regarding e-cigarette use which include frequency, nicotine content and flavours was reported by Morean et al. [48] whilst Oh et al. quantitatively measured frequency of e-cigarette via multiple options question [40]. The classification of e-cigarette measurement in human studies highlighted in this review is summarised in Table 1.

Cell lines are commonly used to investigate the biological effects of e-cigarette liquid which include cells derived from lungs such as bronchial epithelial cells [56–58], alveolar macrophages [59], alveolar type II cells [60] and other cells such as osteosarcoma [61] and human monocyte-macrophage cell line [62]. To date, there is only one study that measured the effects of e-cigarette on adipocytes which was highlighted in this study. The method in in vitro exposure of e-cigarette was established using enriched cell culture medium. The authors accomplished this by using two impingers connected to the custom made smoking machine that is able to control the puff numbers, duration as well as the volume. The aerosol obtained from this machine will be dissolved in the culture medium generating a condensed liquid from each aerosol [51].

## Association between e-cigarette and bodyweight changes

There are two studies which involve in vitro and human subjects that showed no association between e-cigarette and body weight changes. The main aim of the in vitro study is to measure the effects of cigarette and alternative smoking/vaping products on pre-adipocyte differentiation towards beige adipocytes [51]. Adipocyte differentiation is characterised by transcriptional changes of specific gene which result in phenotype alteration in the pre-adipocytes [63]. The differentiation of 3T3-L1 cells to beige adipocytes was measured by the changes in the morphological changes indicated by Oil Red O staining and the expression of brown and beige-selective differentiation markers which were determined by RTPCR. The authors reported e-cigarette did not significantly affect the induction of differentiation to beige adipocytes despite being treated with the same nicotine content as conventional cigarette smoking. Interestingly, *Resistin* which is a general adipogenic marker was significantly increased upon e-cigarette treatment. However, no changes were observed on the mRNA levels on brown- and beige-selective marker, *Pgc-1α*. The authors suggest there are different metabolic effects of e-cigarette that are independent of nicotine and concluded that e-cigarette has limited or no adverse effects on cell survival and the ability to differentiate.

The human studies aim to determine the association between weight control behaviours and current e-cigarette use among school students. This study found no associations between trying to control weight, weight control behaviour and the use of e-cigarette [43]. It was also noted that unhealthy weight control behaviours which include eating less food, fewer calories, taking laxatives, diet pills consumption and fasting for 24 hours or more among adolescents

**Table 1. Comparison of demographic data and e-cigarette consumption measurement in human studies.**

| Author | Pub Year | Source | No of participants | Aged | Time | E-cigarette consumption measurement |
|---|---|---|---|---|---|---|
| [40] | 2020 | Korean National Health and Nutrition Examination Survey | 17656 | > 20 years of age | between the years 2014–2017 | The use of e-cigarette is divided into:<br>1. No- smoker,<br>2. Ex-smoker,<br>3. Current smoker (conventional),<br>4. Current smoker (e-cigarette) |
| [41] | 2020 | 6th Korean National Health and Nutrition Examination Survey | 14738 | > 19 years of age | 2013 to 2015 | The use of e-cigarette is divided into:<br>1. Current e-cigarette users are those who answered yes to the use of e-cigarette in the past as well as for the past 30 days<br>2. Never users are those who answered no to the use of e-cigarette in the past as well as for the past 30 days<br>3. Ever users are those who answered yes to the use of e-cigarette in the past but no to the use of e-cigarette in the past 30 days |
| [43] | 2020 | 2017 Zhejiang Youth Risk Behaviour Survey | 17359 | >13 years old | 2017 | Quantitative analysis of the usage of e-cigarette for the past 30 days |
| [46] | 2019 | Cherokee Nation Health Services primary care outpatient facility near tribal headquarters in Tahlequah | 375 | >18 years old | | The use of e-cigarette is divided into:<br>1. Never user who is defined as never used or tried e-cigarette before<br>2. Ever user who is defined as used and tried e-cigarette once or twice |
| [42] | 2018 | Two 4-year and four 2-year colleges belonging to single university system in Oahu, Hawaii. | 470 | 18–25 years age range | Fall of 2016 and spring of 2017 | The use of e-cigarette is divided into:<br>1. Current users are those who identified themselves or used e-cigarette in the past 30 days<br>2. Experimental users are those who currently non-smoker and had smoked less than 100 cigarettes in the past |
| [44] | 2018 | Texas adolescent Tobacco and Marketing Surveillance System (TATAMS) | 2733 | 12–19 years old | November 2015-January 2016 | The use of e-cigarette is divided into either ever tried e-cigarette or the use of e-cigarette for the past 30 days |
| [45] | 2018 | 2015 National Risk Behaviour Survey | 15,129 | 15–18 years old | 2015 | The use of e-cigarette is divided into:<br>1. Current user who is defined as respondent that used e-cigarette for the past 30 days<br>2. Dual user who is defined as respondent that used both conventional and e-cig for the past 30 days |
| [47] | 2017 | Undergraduate attending California State University, Long Beach | 452 | >18 years old | 2015–2016 | The use of e-cigarette is divided into:<br>1. Ever smoked e-cigarette users<br>2. Individual perception as a regular user |
| [48] | 2016 | Amazon Mechanical Turk | 459 | >18 years old | Sept 2015 | Measured: Duration of e-cigarette, flavour, the absence or presence of nicotine content in the liquid |
| [39] | 2015 | 2011 California Longitudinal Smoker Survey who participated in 2009 California Health Interview Survey | 1000 | >18 years old | July 2011 and concluded in April 2012 | The use of e-cigarette is divided into:<br>1. Experimented users if they have used e-cigarette in the past<br>2. Not experimented users are all other responses (might/never) |

were more likely to be current e-cigarette users. This suggests weight control is not one of the main factors that promote the use of e-cigarette among Chinese adolescents.

Weight control behaviour among e-cigarette was also investigated in a study using online survey conducted to identify adult e-cigarette users that use e-cigarette for weight loss or weight control [48]. The study reported frequent e-cigarette users who are currently over-weight and engaged in calorie restriction to manage body weight were more likely to use e-

cigarette to lose or control body weight. Vanilla and coffee-flavoured e-liquids are among the flavours that were significantly associated with e-cigarette used to control body weight. Nicotine content, cigarette smoking status and unhealthy behaviours such as binge eating, using laxatives and vomiting were not associated with e-cigarette use.

Three human studies in this review investigated the association between perceived weight status and e-cigarette use. The study among young adult college students located on the island of Oahu in Hawaii reported higher weight concerns were significantly associated with higher likelihood of current conventional cigarette smoking status, frequencies and smoking susceptibilities [42]. Interestingly, no association was noted between weight concerns and e-cigarette use status and susceptibility. However, higher weight concerns were associated with higher current e-cigarette use frequency. A study among the American Indian population in one of the primary care outpatient facility reported no significant association between sex, age group or education regarding weight control perception and e-cigarette use [46]. Nonetheless, respondents that used e-cigarette more than once in the past has significantly higher belief that e-cigarette use helps to keep weight down. A larger sample size study of adolescents conducted by Centre for Disease Control and Prevention (CDC) showed a significant association between perceived weight status and e-cigarette use in female adolescents [45]. Female adolescents who perceived themselves as overweight were more likely to report as current e-cigarette users compared to those who perceived themselves as normal weight. Highest prevalence of e-cigarette was reported in males and females who are overweight. In conclusion, all three studies showed higher weight concern is associated in higher e-cigarette use.

Experimentation of e-cigarette among conventional smokers was reported in a longitudinal study conducted to examine the changes observed in reports of e-cigarette experimentation in high-risk sub groups which include smokers who are either obese or overweight [39]. A greater increase in experimentation of e-cigarette was noted in female smokers who are obese and overweight. On the other hand, the study reported neither the expectancies that smoking influence body weight nor concern regarding weight gain were related to the experimentation of e-cigarette use. The results from this study postulates that the increase in e-cigarette experimentation has no significant effect on tobacco use or cessation rate among smokers who are either obese or overweight.

A cross sectional study conducted among undergraduate students to determine the association between weight status and e-cigarette use [47] reported respondents who are obese are more likely to use e-cigarette compared to alcohol use. The study showed weight status was associated with e-cigarette use. Similar findings were reported in a cross sectional study among adolescent to determine the association between weight status and the use of e-cigarette for the past 30 days. A positive association was found in males, but not for females. It was noted that males with obesity had increased odds of past 30-day e-cigarette use compared to their peers with healthy weight [44].

Two human studies examined the association between e-cigarette and metabolic syndrome using data from the Korean National Health and Nutrition Examination Survey. The first study which used data from 2013 to 2015 reported a greater waist circumference and higher triglyceride levels in current e-cigarette male users compared to male that has never used e-cigarette [41]. E-cigarette use was also significantly associated with an increased odds ratio for metabolic syndrome. The second study which used data from the same survey but conducted between 2014 and 2017 among adults reported similar findings [40]. Prevalence of metabolic syndrome in females was significantly associated with cigarette type with both conventional and e-cigarette had increased odds for metabolic syndrome. The cigarette type in females was associated with components of metabolic syndrome with both conventional and e-cigarette had increased odds for high triglycerides and high fasting plasma glucose. No association

between cigarette type and metabolic syndrome was noted in males. In summary, both studies noted metabolic syndrome is associated with higher e-cigarette use.

Interestingly, two in vivo studies reported a significant weight decrease upon e-cigarette consumption. The main objective of one of the in vivo study was to investigate the effect of e-cigarette on cardiac structure and function in diet-induced obese mice [49]. A significant weight decrease was observed in high fat diet fed mice treated with 2.4% nicotine delivered through e-cigarette compared to control groups which are high fat diet fed mice either treated with normal saline or aerosol with 0% nicotine. An increase in plasma free fatty acid levels was observed in mice on high fat diet expose to e-cigarette. Transmission electron microscopy analysis on the same group indicated lipid accumulation at ventricular intramyocardial or also known as ventricular steatosis.

The study on mated mice exposed to e-cigarettes showed e-cigarette exposure *in utero* results in decreased weight gain in females [50]. The litters aged 8 to 12 weeks from dams exposed to e-cigarette for 4 months were mated. The *in utero* e-cigarette exposure of dams resulted in slight reduction in the F2 offspring weight. The female mice exposed *in utero* to e-cigarette showed visible smaller inguinal fat pads and adipocytes compared with the mice in the sham group. This implies the e-cigarette exposure *in utero* regulates metabolic function in adulthood.

## Discussion

E-cigarettes are primarily used as a smoking cessation device which are viewed as a benefit in reducing nicotine withdrawal symptoms but at the same time can be a hindrance in dealing with addiction [64]. The use of e-cigarette is considered as a healthier option compared to tobacco as they are perceived to be lack of tar and carcinogens [65]. This is most likely the reason it is popular as an alternative method compared to conventional nicotine smoking in reducing body weight among patients with obesity. However, mixed findings were noted on the association between e-cigarette use and weight control.

From 10 human studies analysed in this review, six studies concluded higher prevalence of e-cigarette among respondent who are obese or overweight and two studies reported an increased odds for metabolic syndrome. The association between e-cigarette and weight control is inconclusive with one study showed an increase use of e-cigarette use whilst the other reported no associations. Both of the in vivo studies reported a decrease in bodyweight upon e-cigarette exposure. However, the in vitro study showed no significant effect of e-cigarette on adipocyte differentiation. The findings from the studies highlighted in this review are summarised in Table 2.

The lack of information regarding the content of e-cigarette results in inconclusive evidence on the effects of e-cigarette towards the human body. Insufficient data about the long-term health impact of e-cigarette use and given that the majority of e-cigarette users are ex-smokers, and thus disaggregating the effect of e-cigarette use from previous smoking history is challenging [66]. The majority of the cross sectional studies in this review used secondary data to measure either the general health status or to monitor health risk behaviour of the respondents and the information regarding e-cigarette was extracted from these surveys. Hence, detailed information regarding e-cigarette consumption is not available for analysis.

To measure the effect of e-cigarette on body weight changes, it is important to identify the confounding variables that aid the accurate measurement of e-cigarette exposure. Several human studies in this review did not address confounding factors regarding e-cigarette consumption which includes type of e-cigarette, nicotine content of the e-cigarette consumed, type of flavours or whether the respondents are dual users [39–42, 45, 48]. These factors are associated with e-cigarette dependence alongside exposure.

**Table 2. Characteristics of relevant studies.**

| | Author | Type of study | Results | Limitations | Conclusions |
|---|---|---|---|---|---|
| 1. | [40] | Cross sectional | 1. No association between cigarette type and metabolic syndrome.<br>2. Prevalence of metabolic syndrome was significantly associated with cigarette type in female.<br>3. No association between smoking history and metabolic syndrome in female.<br>4. Cigarette type is associated with component of metabolic syndrome in female. | 1. Respondents were not asked about e-cigarette history, hence<br>• exclusive ex-e-cigarette smokers could not be separated from ex-conventional smokers.<br>• history or duration of e-cigarette can be adjusted for in pack in years.<br>2. Study is limited to adults.<br>3. No causal inferences can be made.<br>4. The risks of metabolic syndrome persist to 20 years after cessation. Hence, it is impossible to identify the onset of metabolic syndrome. | Metabolic syndrome is associated with higher e-cigarette use in female. |
| 2. | [41] | Cross sectional | 1. Waist circumference was greater in current e-cigarette male user than male never e-cigarette user.<br>2. Triglyceride levels were the highest in the current male e-cig users.<br>3. Current e-cigarette user showed significantly higher odds ratio for abdominal obesity and hypertriglyceridemia than never e-cig user.<br>4. Dual e-cigarette and cigarettes users were associated with increased odds ratio for abdominal obesity.<br>5. E-cigarette use was significantly associated with increased odds ratio for metabolic syndrome.<br>6. Odds ratio for metabolic syndrome was the highest in current e-cigarette user and the lowest in never e-cigarette user. | 1. No causal inferences can be made.<br>2. No data on e-cigarette such as types of e-cigarette, flavours, consumption patterns.<br>3. No smoking history for former smokers. | E-cigarette use was significantly associated with an increased odds ratio for metabolic syndrome. |
| 3. | [43] | Cross sectional | Unhealthy weight control behaviours which include eating less food, fewer calories, taking laxatives, diet pills consumption and fasting for 24 hours or more among adolescents were more likely to be current e-cigarette users. | 1. Data is self-reported.<br>2. No causal inferences can be made. | No association between weight control behaviours and current e-cigarette use. |
| 4. | [46] | Cross sectional | Respondents that used e-cigarette more than once in the past has significantly higher belief that e-cigarette use helps to keep weight down. | 1. Small sample size.<br>2. Convenience sampling was used.<br>3. No assessment of BMI. | E-cigarette users believe that e-cigarette use help to keep weight down. |
| 5. | [42] | Cross sectional | 1. Weight concerns are statistically significantly associated with increased likelihood of having ever experimented with cigarettes.<br>2. Higher weight concerns were associated with higher frequency of current e- cigarette use. | 1. The use of sample population 18–25 years- may not generalize to older or younger populations.<br>2. No causal inferences can be made.<br>3. No association can be made between dual users due to small sample size. | Higher weight concerns were associated with higher e-cigarette use frequency. |
| 6. | [44] | Cross sectional | 1. Overweight youth did not have increased odds of ever or past 30-day cigarette or e-cigarette use compared to healthy weight youth.<br>2. Increased odd for past 30-day e-cigarette use in male who are obese compared with healthy weight. | 1. Some analyses are underpowered due to low prevalence.<br>2. BMI data is self-reported.<br>3. Limited generalization. | Obese respondents had increased odds of e-cigarette use. |
| 7. | [45] | Cross sectional | 1. Overweight group showed the highest prevalence of e-cigarette use.<br>2. Female adolescents who perceived themselves as overweight were more likely than those perceived themself as normal weight to report as current e-cigarette users. | 1. Data is self-reported.<br>2. The data does not distinguish nicotine-free e-cigarette users.<br>3. Data of absent and dropout were excluded even though these students have more high risk behaviour.<br>4. No causal inferences can be made. | Highest prevalence of e-cigarette was reported in respondents who are overweight. |
| 8. | [47] | Cross sectional | Respondents who are obese and deviated from normal BMI had higher likelihood of using e-cigarette compared to the high substance. | 1. Data is self-reported.<br>2. Limited generalization.<br>3. No causal inferences can be made. | Weight status was associated with e-cigarette use. |

(*Continued*)

**Table 2.** (Continued)

|  | Author | Type of study | Results | Limitations | Conclusions |
|---|---|---|---|---|---|
| 9. | [48] | Cross sectional | 1. Frequent vapers who currently engaging in calorie restriction as a weight loss strategy were reported being overweight are more likely to report currently vaping to lose/control weight.<br>2. Vanilla and coffee-flavoured e-liquids were significantly associated with vaping to lose/control weight.<br>3. 50% of participant who endorsed vaping for weight management were men<br>4. Unhealthy behaviours such as binge eating, using laxatives and vomiting were not associated with vaping to lose/control weight.<br>5. No difference in vaping to lose/control weight based on nicotine content or cigarette smoking status. | 1. Data is self-reported.<br>2. Limited generalization.<br>3. The eating disorder behaviour was not determined using DSM-5.<br>4. Limited flavours of vape were assessed<br>5. A single motive of weight loss/control was measured.<br>6. No data on thee-cigarette frequency. | E-cigarette is used to control body weight. |
| 10. | [39] | Longitudinal | 1. A significant increase in rates of experimentation with e-cigarette among smokers who are obese or overweight.<br>2. Women are either obese or overweight are significantly higher in experimenting e-cigarette compared to men.<br>3. Experimentation of e-cigarette is not related to neither expectancy that smoking helps to control weight nor weight gain upon quitting. | 1. Data is self-reported<br>2. No details about reasons for experimentation<br>3. Limited generalization.<br>4. No data on the frequency. | A greater increase in experimentation of e-cigarette was noted in smokers who are obese and overweight. |
| 11. | [50] | In vivo | 1. E-cigarette results in decreased weight gain in females.<br>2. Female mice exposed in utero to e-cigarette showed visible smaller inguinal fat pads and adipocytes compared with the mice in the sham group. | Other environmental factors which include male reproductive fitness and genetic factors were not evaluated. | Decreased body weight upon e-cigarette exposure. |
| 12. | [49] | In vivo | 1. Nicotine delivery through e-cigarette significantly decreased bodyweight in high fat diet fed mice compared to control.<br>2. Increase in free fatty acid in the plasma suggested to increased lipolysis. |  | A significant weight decrease upon e-cigarette consumption. |
| 12. | [51] | In vitro | 1. No impairment of 3T3-L1 cell survival was observed with e-cigarette.<br>2. Significant increase in Resistin (general adipogenic marker) expression compared to control. | 1. The experiment was not extended in using a negative control agent such a nicotine receptor blocker.<br>2. During cell differentiation, no difference was observed between control and treatment group.<br>3. The use of in vitro study and clinical relevance. | E-cigarette has limited or no adverse effects on adipogenic cell differentiation. |

One of the key focuses in e-cigarette marketing is the flavourings in e-cigarette fluids and there are more than 7764 unique flavour names available in the market [67]. The flavours of e-cigarette were shown to increase product appeal and enjoyment among adults [68] whilst in younger adults, they are associated with vaping initiation and satisfaction[69]. E-cigarettes comes in various nicotine concentration ranging from 0–40 mg/ml [70]. Some studies have attempted to quantitatively measure the e-cigarette use consumption by calculating the amount of cartridges used; each cartridge contains 7.25 mg nicotine [71] or the amount of nicotine concentration in the consumed liquid [72]. Other studies have compared e-cigarette use with tobacco cigarette to measure its consumption for example 10 puffs on e-cigarette is equivalent

to one tobacco cigarette [73]. It was also noted that the different models of e-cigarette was associated with the efficiency in nicotine delivery which may contribute to dependence [74].

Rodents are commonly used as an animal model for practical and economics reasons. Furthermore, the effects in the offspring can be observed within a short time compared to other animal models. The in vivo study investigating in utero effects of e-cigarette highlighted few limitations in their study which involve measuring other confounding factors as well as the fitness level of the reproductive system in male mice [50]. Obese maternal factors in mice that might contribute independently to the phenotype of the offspring include the presence glucose intolerance, diabetes and insulin resistance. Thus, developmental programming can be confounded by the presence of more than one insult [75]. Other confounding factor that might contribute to intrauterine growth restriction which eventually lead to reduced lipid accumulation in offspring is preeclampsia [76]. Absence of direct assessment of these factors may limit the interpretation of data obtained for both maternal and offspring.

Interestingly, there is no common definition of e-cigarette use prevalence noted in the literature. Many studies have adopted the current use of e-cigarette based on e-cigarette used in the past 30 days. In contrast, the term every day or some days without lifetime threshold question is adapted from the conventional cigarette prevalence measurement. Other studies measure e-cigarette current use via multiple choice options for self-reported frequency [77].

To date there are several studies that attempt to measure e-cigarette dependence using modified nicotine or cigarette measures. Penn State Electronic Cigarette Dependence Index (ECDI) comprises ten modified items that captures aspects of e-cigarette dependence which includes data on length of e-cigarette use sessions, the time of the day when e-cigarette was used, craving and withdrawal symptoms [74]. This survey was used in many studies including a study which aims to measure the e-cigarette smoking behaviors and intentions for future cigarette smoking among adolescent [78]. The Patient-Reported Outcomes Measurement Information System (PROMIS) Smoking Initiative has developed six item banks based on item response theory (IRT) to assess cigarette smoking behavior and biopsychosocial constructs associated with current daily and nondaily cigarette smokers. The PROMIS consists of qualitative phase and quantitative analysis in large samples [79]. PROMIS was modified and was used to measure e-cigarette dependence in exclusive e-cigarette users and in dual-users of cigarettes and e-cigarette. The study concluded that this survey showed good psychometric properties, encouraging preliminary validity and the easy access of this survey provided convenient usage [80]. The Population Assessment of Tobacco and Health (PATH) Study is a longitudinal study conducted annually to measure tobacco use and its associated measure. The is another example of study that measured dual user, frequency of cigarette smoking, frequency of e-cigarette use, regular/last e-cigarette flavour use, past 30-day e-cigarette flavour use, and e-cigarette device type use [81].Several interesting findings regarding e-cigarette and smoking cessation were reported from the first and second wave of PATH data [82–85].

One of the in vivo studies included in this review used the SCIREQinExpose whole body inhalation system for 3 hours a day with two puffs per minute whereby each puff lasts for 2 seconds. The vapour solution contains 24ng/ml nicotine [50]. SCIREQinExpose is a computerised inhalation machine designed to control both exposure period and duration of e-cigarette as well as tobacco cigarette. This machine requires rodents to be restrained and commonly, only the nose will be exposed to the treatment [86]. The authors in the other in vivo study developed their own e-cigarette aerosol exposure system named EcigAero. The male mice was exposed to 2.4% nicotine for 12 hours daily for 12 weeks [49]. EcigAero (3 eCig mouse model) allows controlled e-cigarette aerosol delivery to mobile rodents via inhalation to measure e-cigarette effect in vivo. This system includes an aerosol exposure chamber that can house 5 unrestrained mice and connected to a pressurized air source which allows appropriate flow of e-

cigarette aerosol. The e-cigarette aerosol concentration, air pressure, air-flow rate as well as puff duration and frequency of puffs are controlled by e-cigarette activation control unit [87].

As a conclusion, high prevalence of e-cigarette was noted among obese population. Nonetheless no causal inference can be concluded as majority of the human studies were cross sectional. More conclusive findings were identified in in vivo studies with e-cigarette causing weight decrease. However, these observations were not supported by in vitro data. Hence, the effect of e-cigarette on body weight changes warrants further investigations.

Future research should utilise questionnaires that take into account e-cigarettes that do not contain nicotine and e-cigarettes that contain other addictive substances, smoking history, the frequency and the duration of e-cigarette used in a day as well as the flavours of e-cigarette. The gaps in understanding the molecular and cellular mechanisms of e-cigarette exposure and reduced lipid accumulation may involve several diverse pathways such as energy metabolism, dysregulation of inflammatory cells and immune processes. Hence in vitro and vivo studies exploring these pathways will add more wealth and conclusive findings to the current limited knowledge about the effect of e-cigarettes.

## Supporting information

**S1 Table. Risk of bias assessment of animal and human studies according to criteria defined in the OHAT risk of bias tool for human and animal studies.**
(DOCX)

**S1 Checklist. PRISMA 2020 checklist.**
(DOCX)

## Author Contributions

**Conceptualization:** Sandra Maniam.

**Data curation:** Rafidah Hod, Nurul Huda Mohd Nor, Sandra Maniam.

**Formal analysis:** Rafidah Hod, Nurul Huda Mohd Nor, Sandra Maniam.

**Methodology:** Rafidah Hod, Nurul Huda Mohd Nor, Sandra Maniam.

**Project administration:** Nurul Huda Mohd Nor, Sandra Maniam.

**Validation:** Sandra Maniam.

**Writing – original draft:** Rafidah Hod, Sandra Maniam.

**Writing – review & editing:** Rafidah Hod, Nurul Huda Mohd Nor, Sandra Maniam.

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
