## [Decision Letter · Decision Letter 0]

26 May 2022

PONE-D-21-37846Systematic Review on E-Cigarette and its Effects on Weight Gain and AdipocytesLOS ONE

Dear Dr. Maniam,

Thank you for submitting your manuscript to PLOS ONE. After careful consideration, we feel that it has merit but does not fully meet PLOS ONE’s publication criteria as it currently stands. Therefore, we invite you to submit a revised version of the manuscript that addresses the points raised during the review process.

We look forward to receiving your revised manuscript.

Kind regards,

Luca Vanella

Academic Editor

PLOS ONE

Journal Requirements:

Important: If there are ethical or legal restrictions to sharing your data publicly, please explain these restrictions in detail. Please see our guidelines for more information on what we consider unacceptable restrictions to publicly sharing data: http://journals.plos.org/plosone/s/data-availability#loc-unacceptable-data-access-restrictions. Note that it is not acceptable for the authors to be the sole named individuals responsible for ensuring data access

Reviewers' comments:

Reviewer's Responses to Questions

**Comments to the Author**

1. Is the manuscript technically sound, and do the data support the conclusions?

Reviewer #1: Yes

2. Has the statistical analysis been performed appropriately and rigorously? 

Reviewer #1: Yes

3. Have the authors made all data underlying the findings in their manuscript fully available?

Reviewer #1: Yes

4. Is the manuscript presented in an intelligible fashion and written in standard English?

Reviewer #1: Yes

5. Review Comments to the Author

Reviewer #1: As befits a correct Systematic Review, the authors summarized an important series of data concerning the link between E-Cigarettes and the weight gain, highlighting several aspects relatively unexplored.

In particular, I really appreciated the presence of the two tables and the flow chart, in order to better clarify every point and to explain the ratio behind the choice of the analyzed articles.

Only few suggestions, useful to refine an actual high-level review.

1. As I said, I appreciate when the tables in reviews can make the data clearer. In this case they do, but I noticed some little issues with the table formatting (especially table 1). Can you avoid the “jumping” words? That also in order to define the limits of every virtual column and avoid misunderstandings. Moreover, I would prefer if the author could choose an order for the points of the table 1 (for example, listing the papers following the year of publication). I suggest also to check every possible typing error (missing some capital letters).

2. The attention to details makes happier every reviewer: please, check the numbers at the bottom of the page.

3. To better understand the difference between E-cigarettes (or other similar devices) and classic cigarettes, and how they could cause a different toxicity, I suggest to check a really recent multi-site study (Replica) conducted by the CoEHAR (Centre of Excellence for the Acceleration of Harm Reduction). It could be a nice reading, useful to enrich an already well written introduction.

6. PLOS authors have the option to publish the peer review history of their article (what does this mean?). If published, this will include your full peer review and any attached files.

Reviewer #1: No

---

## [Author Response · Author response to Decision Letter 0]

31 May 2022

Comments Response 

As I said, I appreciate when the tables in reviews can make the data clearer. In this case they do, but I noticed some little issues with the table formatting (especially table 1).

a) Can you avoid the “jumping” words? That also in order to define the limits of every virtual column and avoid misunderstandings.

 R: We have formatted the columns to prevent ‘jumping’ of words. (Table 1 Pg 10-12, Table 2 Pg 19-24)

b) Moreover, I would prefer if the author could choose an order for the points of the table 1 (for example, listing the papers following the year of publication). 

 R: Table 1 was rearranged according to the publication year in descending order. Table 2 was rearranged according to type of study.

c) I suggest also to check every possible typing error (missing some capital letters).

 R: Thank you for the comments. We have checked and corrected the possible typos.

The attention to details makes happier every reviewer: please, check the numbers at the bottom of the page.

 R: We apologise for the mistake and have corrected the page numbers accordingly.

To better understand the difference between E-cigarettes (or other similar devices) and classic cigarettes, and how they could cause a different toxicity, I suggest to check a really recent multi-site study (Replica) conducted by the CoEHAR (Centre of Excellence for the Acceleration of Harm Reduction). It could be a nice reading, useful to enrich an already well written introduction.

R: Thank you very much for the comment. We have improved our introduction and added a couple of references (Pg 4)

---

## [Editor Report · Decision Letter 1]

21 Jun 2022

Systematic Review on E-Cigarette and its Effects on Weight Gain and Adipocytes

PONE-D-21-37846R1

Dear Dr. Maniam,

We’re pleased to inform you that your manuscript has been judged scientifically suitable for publication and will be formally accepted for publication once it meets all outstanding technical requirements.

Kind regards,

Luca Vanella

Academic Editor

PLOS ONE
---

## [Editor Report · Acceptance letter]

23 Jun 2022

PONE-D-21-37846R1 

Systematic Review on E-Cigarette and its Effects on Weight Gain and Adipocytes 

Dear Dr. Maniam:

I'm pleased to inform you that your manuscript has been deemed suitable for publication in PLOS ONE. Congratulations! Your manuscript is now with our production department. 

Kind regards, 

on behalf of

Dr Luca Vanella 

Academic Editor

PLOS ONE